# A Pilot Study of Training Peer Recovery Specialists in Behavioral Activation in the United States: Preliminary Outcomes and Predictors of Competence

**DOI:** 10.3390/ijerph20053902

**Published:** 2023-02-22

**Authors:** Morgan S. Anvari, Mary B. Kleinman, Dwayne Dean, Alexandra L. Rose, Valerie D. Bradley, Abigail C. Hines, Tolulope M. Abidogun, Julia W. Felton, Jessica F. Magidson

**Affiliations:** 1Department of Psychology, University of Maryland, College Park, College Park, MD 20742, USA; 2Center for Health Policy & Health Services Research, Henry Ford Health System, Detroit, MI 48202, USA

**Keywords:** task sharing, peer recovery, behavioral activation, training

## Abstract

Background: The peer recovery specialist (PRS) workforce has rapidly expanded to increase access to substance-use disorder services for underserved communities. PRSs are not typically trained in evidence-based interventions (EBIs) outside of motivational interviewing, although evidence demonstrates the feasibility of PRS delivery of certain EBIs, such as a brief behavioral intervention, behavioral activation. However, characteristics that predict PRS competency in delivering EBIs such as behavioral activation remain unknown, and are critical for PRS selection, training, and supervision if the PRS role is expanded. This study aimed to explore the outcomes of a brief PRS training period in behavioral activation and identify predictors of competence. Method: Twenty PRSs in the United States completed a two-hour training on PRS-delivered behavioral activation. Participants completed baseline and post-training assessments, including roleplay and assessments of PRS characteristics, attitudes towards EBIs, and theoretically relevant personality constructs. Roleplays were coded for competence (behavioral activation specific and PRS skills more broadly, i.e., PRS competence) and changes were assessed from baseline to post-training. Linear regression models tested factors predicting post-training competence, controlling for baseline competence. Results: There was a significant pre-post increase in behavioral activation competence (*t* = −7.02, *p* < 0.001). Years working as a PRS significantly predicted post-training behavioral activation skills (*B* = 0.16, *p =* 0.005). No variables predicted post-training PRS competence. Conclusions: This study provides preliminary evidence that behavioral activation may be appropriate for dissemination to PRSs through brief trainings, particularly for PRSs with more work experience. However, additional research is needed to examine predictors of competence among PRSs.

## 1. Introduction

In 2020 alone, 37.1 million people in the U.S. needed, yet did not receive, substance use disorder (SUD) treatment [1]. This treatment gap has disproportionately impacted underserved, ethnic/racial minoritized populations compared to non-racial/ethnically minoritized populations [2,3]. To increase access to care in low-resource, underserved communities, the workforce of peer recovery specialists (PRSs; also known as peer recovery coaches or peer recovery advocates dependent on location), individuals with lived substance use and recovery experience, has rapidly expanded [4]. PRSs provide a variety of services, such as linkage to resources, case management, and assistance in navigating the health care systems. PRS services have been found to be linked to reduced substance use/return to use, improved relationships with treatment providers, increased treatment retention, and greater treatment satisfaction [5,6]. PRSs are often trained in motivational interviewing; however, they are not consistently trained in the delivery of any other evidence-based interventions (EBIs) [4,5,7,8].

One EBI that may be promising for PRS delivery is behavioral activation (BA). BA is a manualized EBI initially developed for the treatment of depression, which has since been adapted for, and shown efficacy in, the treatment of SUD [9,10,11]. BA is typically delivered as a structured, six- to twelve-session intervention; however, the core principles of BA may be integrated with other treatment approaches. BA for SUD often includes content on: (1) identifying the cycle of substance use (i.e., negative feelings, urges and behaviors); (2) breaking the negative cycle; and (3) identifying and scheduling substance-free activities that are in line with one’s values. Emerging evidence led by our team has demonstrated that PRSs can deliver BA with feasibility, acceptability, and fidelity, and that peer-delivered BA has been associated with improvements in SUD outcomes [6,10,12,13,14]. However, there is a lack of research regarding best practices (i.e., length, teaching techniques, etc.) for training PRSs in EBIs to inform widespread dissemination and predictors of successful training of EBIs, such as BA, among the PRS workforce.

While evolving evidence supports the fact that BA can be feasibly and effectively delivered by PRSs, results (described above) are drawn from a small sample of PRSs specifically hired for intervention studies. Across types of interventions, factors such as predictors of competence, patient outcomes, and working/therapeutic alliance among traditional mental health care providers (i.e., counselors, therapists, etc.) are mixed. A number of studies have examined these outcomes among mental healthcare providers delivering traditional psychotherapy, yielding conflicting results on how variables, such as personality type, level of experience, and level of training, relate to positive outcomes [15,16,17,18,19].

To our knowledge, literature examining these relations within the delivery of BA is non-existent, even among clinicians. There is, however, evidence suggesting that clinician personality factors may relate to patient outcomes when receiving cognitive behavioral therapy (CBT). In one study, patients who received CBT from clinicians with above-average openness to experience scores on a commonly used personality measure had poorer treatment outcomes than those working with therapists with average/lower openness to experience scores [20].

While results suggesting the relation between personality attributes and clinician effectiveness may point to potential predictors of competence (i.e., fidelity) in PRSs, there is no research, to our knowledge, that may distinguish characteristics predictive of learning and delivering BA skillfully among PRSs. As this workforce is expanded potentially to include the delivery of EBIs, it is important to explore factors (e.g., personality constructs, years of experience, etc.) that are associated with the faithful delivery of EBIs by PRSs. Some PRSs may feel that delivering a manualized EBI is not within their role or perceive EBIs as better suited to be delivered by clinicians. Additionally, it is possible that certain characteristics of PRSs or their relationship with their clients may impact the fidelity of delivery of BA. For instance, differences between PRS and client recovery pathways may act as a barrier to delivering BA as intended, as PRSs may inadvertently impose their own recovery path onto clients [21].

More guidance is needed in identifying PRS strengths that are associated with delivering EBIs, such as BA. Moreover, as the PRS workforce is expanding, a greater understanding of factors associated with efficient training and delivery of these types of interventions will be vital for widespread dissemination of EBIs. Thus, this study aimed to: (1) evaluate changes in BA and PRS competency skills following a brief training period; and (2) examine factors that may predict PRS BA competence. We predicted that lower levels of stigma towards substance use and medication for opioid use disorder (MOUD), belief in EBIs, high openness and conscientiousness, and being more accepting of multiple pathways to recovery (i.e., low orientation towards abstinence-only) would predict better post-training performance for both BA and PRS competence.

## 2. Method

### 2.1. Participants

Participants (*N* = 20) were either certified PRSs or individuals seeking/working towards PRS certification. Of note, PRS certification requirements in the U.S. vary by state, and may include: years/time in recovery, service hours, hours spent being supervised, examination(s), and ongoing continued education credits [22]. Many states utilize the Connecticut Community for Addiction Recovery (CCAR)-approved training materials, which often include domains such as: ethics, advocacy, mentoring, and recovery/wellness [22]. Recruitment of peers was conducted through PRS network listservs, social media pages, and word-of-mouth led by a PRS on our team. There were no exclusion criteria regarding geographic location. Participants were from various states in the U.S., primarily (though not limited to) the DC-Maryland-Virginia area, where the research team is located. A research assistant (RA) contacted all interested participants to confirm eligibility and schedule assessments and training. While most PRSs were SUD-focused peers, a small subset of them were mental health peers (*n* = 2). Mental health peers are individuals with lived experience with mental health disorders, who receive similar training and certification as SUD-focused PRSs.

### 2.2. Procedures

#### 2.2.1. Training

Trainings were held virtually using Zoom, in a group format with two-to-five participants per training. Trainings were delivered by a certified PRS with supervisory and training credentials as a PRS and training in BA, and a clinical psychology doctoral student with BA expertise. Each training lasted approximately two hours, and began with an overview of basic PRS competencies, including: rapport building, non-verbal and verbal communication skills, disclosure, supporting self-efficacy, and collaborative goal setting. The remaining half of the training focused on fostering proficiency in the core skills of BA for SUD: (1) understanding the cycle of substance use including identifying negative feelings, urges and behaviors; (2) breaking the cycle (i.e., changing behavior) to produce positive outcomes; (3) discussing and identifying life values; and (4) exploring/identifying positive, rewarding activities that are in alignment with one’s life values. The training included interactive discussions, breakout rooms and roleplays to foster skill development.

#### 2.2.2. Assessments

Assessments were completed before and after the training (within two weeks for each). Assessments included both written quantitative measures (detailed below) completed independently by participants (with RA assistance if needed) using an online data collection tool, REDCap (https://www.project-redcap.org/, Version: 10.0.33, accessed on 24 October 2022), as well as roleplay with a trained RA. Participants were provided with a $25 gift card for each assessment and a certificate of completion, which individuals seeking PRS certification were able to use to verify two service hours towards their certification requirements.

Each roleplay was recorded with participant consent, lasted approximately 10–15 min and utilized a trained RA actor. Participants were provided with a brief background on the patient and roleplay instructions (written and verbally) at both assessment points. Mock patient backgrounds varied slightly from the baseline to post-training assessment to avoid recall effects, but presented similar content including SUD, receiving MOUD, and referral through a community outreach program. Participants were instructed that they would be roleplaying as a PRS working with individuals with substance use disorder and receive methadone, and that they had ten min to learn about the person and their experiences and engage in a supportive manner. Participants were encouraged to spend some of their time sharing a bit about their own experience in recovery if they were comfortable. Additional instruction was provided at the post-training assessment to discuss life values and identify activities at post-treatment; this was not provided at baseline under the assumption that participants had not received prior BA training; all participants self-reported prior trainings at baseline assessment and no participants reported previous BA training.

RAs included post-baccalaureate, master’s, and doctoral-level student researchers. RAs received in-depth patient vignettes and approximately three hours of roleplay training, including practice roleplay and discussing how to respond to different scenarios. For example, RAs were asked to respond to closed-ended questions with “yes/no” responses, ask for clarifying information when high-level clinical jargon was used, and not disclose information from the vignette unless specifically probed. RAs were given instruction on non-verbal communication to portray throughout the roleplay, such as speaking softly, sighing, and looking down when speaking about their problems. Further, RAs were instructed to raise specific concerns throughout the course of the roleplay, including (though not limited to): asking if meeting with the PRS will help treat their depression, and that they want to discontinue their methadone medication.

All procedures were approved by the University of Maryland, College Park Institutional Review Board and all participants provided verbal consent prior to study participation. Data were collected between December 2021–February 2022.

### 2.3. Measures

#### 2.3.1. The Evidence-Based Practice Attitude Scale-36 (EBPAS-36)

Attitudes towards EBIs were assessed utilizing the EBPAS-36 [23]. This scale assesses attitudes towards EBIs broadly (i.e., not BA specifically). Higher total scores indicate higher acceptance and positive attitudes towards EBIs.

#### 2.3.2. Substance Use Stigma

Participant experiences of internalized (i.e., negative feelings towards themselves), enacted (i.e., experienced discrimination) and anticipated (i.e., expected future discriminative experiences) stigmas were assessed utilizing the Substance Use Stigma Mechanisms Scale (SU-SMS; [24]) If participants did not self-report substance use (individuals can be a PRS based on shared mental illness diagnoses and/or having a family member or close friend with SUD) in a brief demographic questionnaire, the measure was not administered. Higher subscale and total scores indicate higher levels of stigma.

#### 2.3.3. The Abstinence Orientation Scale (AOS)

Openness to various pathways to recovery, namely MOUD, was assessed using the AOS [25]. Higher total scores indicate stronger negative views towards MOUD.

#### 2.3.4. Social Distance Scale (SDS)

The SDS [26] measures varying degrees of closeness (i.e., warmth, hostility, indifference, or intimacy) in participants towards members of diverse social, ethnic or racial groups. To administer the SDS, researchers developed a vignette describing an individual who portrays characteristics of interest. Participants then indicate how likely they would or would not be to engage in certain behaviors/activities with the individual in the vignette. This study included two vignettes describing two different levels of recovery. The first vignette portrayed an individual who: is actively receiving MOUD; has decreased their use of heroin and cocaine but is still in intermittent periods of active use; has a goal of working towards gaining one week of take-home doses. The second vignette portrayed an individual who: is actively using heroin and fentanyl; does not have interest in reducing or stopping their use; and has entered and discontinued inpatient treatment various times. A total score was computed by adding together the two vignette’s scores in order to indicate total levels of desired social distance to the portrayed individuals. Higher total scores indicate greater preference for distance.

#### 2.3.5. International Personality Item Pool NEO-60 (IPIP-NEO-60)

Openness and conscientiousness, personality traits that have been associated with positive therapist-client outcomes [18,20], were assessed using the respective IPIP-NEO-60 subscales [27]. Higher subscale scores indicate stronger endorsement of the trait.

#### 2.3.6. ENhancing Assessment of Common Therapeutic Factors (ENACT)

The ENhancing Assessment of Common Therapeutic factors (ENACT) scale [28] is a 15-item measure developed to assess competence among non-specialists delivering mental health interventions. In consultation with an experienced PRS on the research team, we selected ENACT items that were most relevant to basic PRS competencies: non-verbal communication and active listening; verbal communication skills; rapport building and self-disclosure; demonstration of empathy, warmth, and genuineness; collaborative goal setting and addressing client’s expectations; promotion of realistic hope for change; and incorporation of coping mechanisms and prior solutions. The ENACT is scored on a scale of 1–4 (1 = harmful; 2–3 = some or all basic skills; 4 = advanced skills). The ENACT provides a detailed codebook defining each score for each skill; the research team and PRS reviewed the existing codebook and adapted as needed to align with typical expectations of PRSs and their work. In order to assess BA skills, the research team also developed a codebook following the same scoring structure as the original ENACT (i.e., a 1–4 scale) for two skills: identifying/discussing life values, and activity identification. Total scores were created by adding all items for each skill domain, resulting in a total ENACT items score and a total BA score.

### 2.4. Roleplay Coding

The coding team consisted of a post-baccalaureate RA and PRS with experience in delivering BA. Coders received training from a doctoral student familiar with both the ENACT and BA, which included detailed discussion of the codebook and coding practices. Following training, coders met weekly to code roleplay recordings. All items were scored independently; immediately after independent scores were determined, coders discussed and resolved any discrepancies via consensus. Consensus scores were used for analytic purposes. Whereas the ENACT, and consequently the BA, codebook scored items as having either “1 = harmful”, and “2 = some basic skills,” etc., coders noted that there was no available score for instances where participant behaviors/skills were not harmful, yet the participant did not display any basic skills. Applicable instances were tracked during data collection to allow for differentiation between individuals with some basic skills versus individuals with no basic skills yet no harmful behaviors (e.g., takes up space by talking without utilizing basic skills, but causes no harm). Roleplay skills were then ultimately scored such that 0 = harmful; 1 = no basic skills; 2 = some basic skills; 3 = all basic skills; 4 = advanced (see Table 1 for full codebook). Though participants were not instructed to used BA skills at baseline, BA competency was still rated such that it was possible that PRSs utilized skills in alignment with BA (e.g., identifying activities) on their own.

### 2.5. Data Analysis

In order to assess change in BA and ENACT competence (i.e., preliminary outcomes of the training), total BA and ENACT scores at baseline and post-training were compared using a paired *t*-test. To examine potential predictors of post-training BA and ENACT scores (i.e., predictors of competence), linear regression models were run, modeling baseline EBPAS-36, SU-SMS, AOS, and SDS vignette total scores, IPIP-NEO-60 conscientiousness and openness subscale scores, and total years of PRS work experience as predictors. All linear regression models controlled for baseline BA and ENACT scores. Three participants did not complete the SU-SMS due to lack of SUD experience and were not included in analyses using this measure. All participants (*N* = 20) completed all other measures.

## 3. Results

Participants primarily identified as female (70%) and Black/ African American (40%) or White (40%). Average amount of PRS work experience was *M* = 3.09 (*SD* = 4.52) years. Participants worked in a variety of settings/roles, including: community outreach, non-governmental organizations/non-profits (including shelters), departments of health, and public substance use treatment programs (including clinics and hospitals). See Table 1 for more detail. See Table 2 for descriptive data on all baseline and post-training skills.

### 3.1. BA Competence

As expected, following the training, we found a significant baseline (M = 2.4, SD = 0.60) to post (M = 4.30, SD = 145) increase in BA competence (t = −7.02, p < 0.01). More years working as a PRS significantly and positively predicted change in BA competence (B = 0.16, p = 0.005), controlling for baseline BA skills. There was a non-significant trend such that higher levels of conscientiousness and less SU stigma were associated with greater BA competence. No other factors were associated with BA competence. See Table 3 and Table 4 for more detail. 

### 3.2. PRS Competence

There were no significant changes in PRS competence from baseline (*M* = 14.2, *SD* = 5.70) to post-training (*M* = 16.55, *SD* = 4.71) (*t* = −1.54, *p* = 0.14). No factors significantly predicted post-training PRS competence. See Table 3 and Table 4 for more detail. 

## 4. Discussion

Expansion of the PRS workforce has the capacity to play a vital role in narrowing the SUD treatment gap through the delivery of EBIs in addition to usual PRS services. This study aimed to examine if a brief training in core PRS competencies and BA could produce meaningful changes in PRS skills, as well as explore predictors of success. Results demonstrate significant changes in BA skills from baseline to post-training and suggest that years of PRS work experience may predict success in gaining BA skills. No other factors predicted PRS competence or BA skills.

Significant differences between baseline and post-training BA skills suggest that a brief training in BA may be useful in preliminary training of PRSs in core BA skills. BA has been found to be feasible and acceptable for PRS delivery in historically underserved, racial/ethnically minoritized communities to increase engagement in SUD treatment [29], including MT retention [30,31], and PRS-delivered BA has been found to be effective in supporting MT retention [6]. However, these studies utilized a single PRS interventionist, limiting the ability to understand what predicts competence in the delivery of the intervention. Thus, this was a pilot study to begin to understand what predictors may be associated with competence in this approach following a brief training. While feasibility of a briefer training model has yet to be examined in real-world care settings, study results may indicate that a brief BA training has potential to help PRS develop foundational skills in BA, particularly for individuals with greater years of PRS work experience. While results were not statistically significant, there was a trend such that having higher levels of conscientiousness and less SU stigma non-significantly predicted post-training BA competence; future research should explore these relationships in larger sample sizes.

Of note, there were no significant differences in PRS competency skills between baseline and post-training. Results may suggest that these skills, (i.e., empathy, non-verbal and verbal skills, etc.) may be less malleable in a brief training and that a two-hour training was not sufficient for training in these broader counseling skills. Previous work utilizing items from the ENACT scale to rate PRS competency found that PRS competency and adherence to the PRS role (i.e., disclosing shared experience and endorsing high PRS competency) did not affect adherence to BA delivery [12]. No significant predictors of PRS competence were found; future work should continue to explore predictors of change in these skills among larger sample sizes.

### Limitations

Findings must be considered within the context of study limitations. Primarily, it is important to note that a 2-h training session is likely not sufficient in producing long-term skills in an EBI such as BA, and rather, may produce competency in basic, core skills. As this study reported on a small sample size (*N* = 20), due to the preliminary nature of the study, the ability to detect predictors may have been limited. Given that this was a pilot study, future, larger-scale, studies should be conducted to replicate these results and explore additional predictors of post-training success. Of note, no PRSs reported experience taking MOUD, limiting generalizability to those without shared MOUD experience. Two PRSs were mental health peers, which may have also impacted results. Furthermore, this training primarily recruited from the DC-Maryland-Virginia area and may not be generalizable to other geographical locations. As all training occurred online, engagement and skill uptake may have been different if delivered in person. Lastly, long-term retainment of skills was not examined; thus, the results cannot speak to the long-term sustainability of PRS BA delivery following a brief training session without long-term support and/or supervision.

## 5. Conclusions

Training the PRS workforce in EBIs such as BA has the capacity to greatly improve EBI access among individuals with SUD from underserved communities. This study provided preliminary evidence that PRSs may be able to learn to deliver BA with fidelity through very brief training sessions, and that it may be advantageous to target dissemination efforts towards those with more years of experience working as a PRS. Future work with larger sample sizes is needed to explore best training practices, as well as possible predictors of success in BA delivery and broader PRS competence.

## Figures and Tables

**Table 1 ijerph-20-03902-t001:** Roleplay Codebook.

Skill	Harmful (0)	Not Present (1)	Some (2) or All (3) Basic Skills	Advanced Skills (4)
ENACT—Non-verbal communication & active listening	• Engages in other activities (e.g., answers phone, completes paperwork) • Laughs at client • Uses inappropriate facial expressions • Inappropriate physical contact	• No basic skills are used, and no harmful behaviors occur	• Allows for silences • Maintains appropriate eye contact • Maintains open posture (body towards client) • Continuously uses supportive body language (head nod) and utterances (uh huh)	• Varies body language throughout session to match client’s content and expression
ENACT—Verbal communication skills	• Interrupts client • Asks many suggestive or leading closed-ended questions (e.g., You didn’t really want to do that right?) • Corrects client (what you really mean …) or uses accusatory statements (you shouldn’t have said that to your husband) • Culturally and age-inappropriate language and terms	• No basic skills are used and no harmful behaviors occur	• Uses open-ended questions • Summarizing or paraphrasing statements • Allows client to complete statements before responding	• Encourages client to continue explaining (tell me more about …) • Clarifies statements in first person • Matches rhythm to client’s, allowing longer and shorter pauses based on client
ENACT—Rapport building & Self-disclosure	• Dominates session describing a personal experience • Minimizes client’s problem by describing how the helper has dealt with this • Asks unnecessary embarrassing personal questions • Discusses confidential information of other clients	• No basic skills are used, and no harmful behaviors occur	• Introduces self and explains role • Makes casual, informal conversation • Asks for client’s introduction (what the client prefers to be called) • Shares general experience related to the client (about one’s community/region)	• Asks client’s reflection on information that helper has shared • Checks in on client’s comfort (offers seat, preferred language)
ENACT—Demonstration of empathy, warmth, and genuineness	• Critical of client’s concerns • Dismissive of client’s concerns • Helper’s emotional response appears inappropriate, fake, or acting	• No basic skills are used, and no harmful behaviors occur	• Is warm, friendly, and genuine throughout session • Continuously shows concern or care for client (that sounds sad, can you tell me more about it?)	• Asks questions to identify what emotions the client was feeling
ENACT—Collaborative goal setting & addressing client’s expectations	• Tells client their goals can’t be met, but does not give a reason • Gives incorrect, misleading, or unrealistic information about treatment goals • Dictates goals for client (forces goal upon client)	• No basic skills are used, and no harmful behaviors occur	• Asks client about goals (expectations) • Clearly explains how client’s goals and expectations fit with treatment plan • Allows for self-directing	• Prioritizing and modification of treatment plans to fit client goals • Works with client to reframe their goals within scope of treatment plan
ENACT—Promotion of realistic hope for change	• Makes negative statements about client’s doubts (how do you expect to get better if you have no hope?) • Gives unrealistic expectations (everything will be solved) • Provides no hope for change (this problem cannot be solved)	• No basic skills are used, and no harmful behaviors occur	• Explains how client can be hopeful about the possibility of change • Praises client for seeking care	• Solicits and explores client’s doubt about treatment • Shares reasons for hope based on helper’s prior experience or client’s behaviors • Discusses reasons for hope when client is doubtful or dissatisfied
ENACT—Incorporation of coping mechanisms & prior solutions	• Makes negative statements about client’s coping mechanisms (that would never work …) • Encourages harmful coping mechanisms	• No basic skills are used, and no harmful behaviors occur	• Asks client about current or past coping mechanisms • Praises client for positive or safe current or prior solutions	• Encourages continued use of positive coping mechanisms • Reflects on prior unhealthy strategies and brainstorms alternatives with client
BA—Identifying/discussing life values	• Tells client what they should/do value	• No basic skills are used, and no harmful behaviors occur	• Defining values • Discussing clients’ values • Allows for self-direction	• Discussing how activities are in alignment with values • Asking what is important to client, soliciting conversation instead of just a list of values
BA—Activity identification	• Suggesting dangerous activities, something very clearly out of the client’s reach	• No basic skills are used, and no harmful behaviors occur	• Discussing/identifying activities • Peer solicits activities ideas without RA offering ideas/probing for it (ex., what did you use to enjoy doing before you started using)	• Seamless transition from values to activities • Scheduling activities • Helping the client think of feasible ideas/a plan (i.e., the how)

**Table 2 ijerph-20-03902-t002:** Participant demographics.

Variable	Participants (*N* = 20)
	*n* (%)
Race	
Black or African American	8 (40%)
White	8 (40%)
Asian/Indian	1 (5%)
American Indian/Alaska Native	4 (20%)
Latinx	3 (15%)
Gender	
Male	6 (30%)
Female	14 (70%)
Mean age, years (SD)	48.9 (10.47)
Mean years of work as a PRS (SD)	3.087 (4.52)
Peer role focus	
Substance use	18 (90%)
Mental health	2 (10%)

**Table 3 ijerph-20-03902-t003:** Descriptive data on baseline and post-training skills.

Variable	Mean Baseline Score (*SD*)	Mean Post-Training Score (*SD*)
ENACT—Non-verbal communication & active listening	2.70 (0.66)	2.90 (0.31)
ENACT—Verbal communication skills	2.45 (1.10)	1.65 (0.59)
ENACT—Rapport building & Self-disclosure	1.75 (1.29)	2.25 (0.72)
ENACT—Demonstration of empathy, warmth, and genuineness	2.00 (1.03)	2.10 (1.29)
ENACT—Collaborative goal setting & addressing client’s expectations	1.90 (1.02)	2.25 (0.85)
ENACT—Promotion of realistic hope for change	2.00 (1.12)	1.95 (1.23)
ENACT—Incorporation of coping mechanisms & prior solutions	1.35 (0.99)	2.45 (1.00)
BA—Activities	1.35 (0.49)	2.40 (0.75)
BA—Life values	1.05 (0.22)	1.9 (0.97)

**Table 4 ijerph-20-03902-t004:** Predictors of change in BA and clinical competency skills.

Variable	BA Total Score		ENACT Total Score	
	*B* (*SE*)	*p*-Value	*B* (*SE*)	*p*-Value
EBPAS-36	−0.82 (0.90)	0.38	−0.91 (3.66)	0.81
SU-SMS	−0.54 (0.36)	0.15	−2.50 (1.42)	0.10
AOS	0.67 (0.66)	0.33	−1.56 (2.64)	0.56
SDS Total Score	−0.04 (0.03)	0.27	−0.21 (0.14)	0.16
IPIP-NEO-60 Conscientiousness	0.08 (0.04)	0.08	−0.17 (0.19)	0.8
IPIP-NEO-60 Openness	−0.003 (0.05)	0.94	0.09 (0.22)	0.70
Total years of PRS work experience	0.16 (0.05)	0.005	0.39 (0.26)	0.30

Note. All analyses controlled for baseline scores in the respective skill by using baseline scores as predictors.

## Data Availability

This article does not currently have a dataset that has been made available through a data repository.

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
