# Peer review of "A Pilot Study of Training Peer Recovery Specialists in Behavioral Activation in the United States: Preliminary Outcomes and Predictors of Competence"

_ijerph, 2023, doi:10.3390/ijerph20053902_

Round 1

Reviewer 1 Report

The topic reflects the relevancy of the study since the findings can contribute a deeper insight into best practices for training PRSs in EBIs, and predictors of successful training of EBIs, such as BA, among the PRS workforce. Provide a greater understanding of factors associated with efficient training and delivery of interventions which are vital for widespread dissemination of EBIs. Reveal significant differences between baseline and post-training BA skills that suggest a brief training in BA has the potential to help PRSs develop foundation skills in BA, particularly for individuals with greater years of PRS work experience.

What are the outcomes of a brief PRS training, and what are the predictors of PRS BA competence?

The training process could be elaborated further in terms of how interactive discussions were conducted by including types of questions or discussions used. And also, how the role plays are coded, i.e. into specific themes, questions or responses.

Conclusion be improved further by specifying more clearly the preliminary outcome and predictors of competence to recap the study’s title and objectives.

Tables:
A table for role-play coding i.e., based on type of questions, responses, or theme.
A table for a list of questions used in the interactive discussions.

I enjoyed reading your paper. It is prepared with detailed explanations. It will be better if you can relate both theoretical and practical contributions of your study in the discussion part to enhance your research novelty since I believe many PhD students worldwide will read and benefit from your work. Anyway, the paper is ready for publication as it is.

Author Response

The topic reflects the relevancy of the study since the findings can contribute a deeper insight into best practices for training PRSs in EBIs, and predictors of successful training of EBIs, such as BA, among the PRS workforce. Provide a greater understanding of factors associated with efficient training and delivery of interventions which are vital for widespread dissemination of EBIs. Reveal significant differences between baseline and post-training BA skills that suggest a brief training in BA has the potential to help PRSs develop foundation skills in BA, particularly for individuals with greater years of PRS work experience. What are the outcomes of a brief PRS training, and what are the predictors of PRS BA competence?

RESPONSE: Thank you for your review and support of our manuscript. We have responded to your feedback below and believe the manuscript has been strengthened as a result.

  1. The training process could be elaborated further in terms of how interactive discussions were conducted by including types of questions or discussions used. And also, how the role plays are coded, i.e. into specific themes, questions or responses.

RESPONSE: Thank you for this suggestion. We have added more detail on the role play guidelines for both RAs and participants on pages 7-8. Specifically, we say:

“Participants were instructed that they would be role playing as a PRS working with individuals with substance use disorder and receive methadone, and that they had ten minutes to learn about the person and their experiences and engage in a supportive manner. Participants were encouraged to spend some of their time sharing a bit about their own experience in recovery if they were comfortable. (pg. 7)

“RAs were given instruction on non-verbal communication to portray throughout the role play, such as speaking softly, sighing, and looking down when speaking about their problems. Further, RAs were instructed to raise specific concerns throughout the course of the role play, including (though not limited to): asking if meeting with the PRS will help treat their depression, and that they want to discontinue their methadone medication.” (pg. 8)

We have also added more detail regarding the coding process on page 10, including clarification that scores were based on the entirety of each role play. We have now included the full codebook in Table 1.  

  1. Conclusion be improved further by specifying more clearly the preliminary outcome and predictors of competence to recap the study’s title and objectives.

RESPONSE: We agree that mapping the language “preliminary outcomes” and “predictors of competence” onto the analyses/results would increase clarity. We now specify this in the data analytic plan on page 11:

“In order to assess change in BA and ENACT competence (i.e., preliminary outcomes of the training), total BA and ENACT scores at baseline and post-training were compared using a paired t-test. To examine potential predictors of post-training BA and ENACT scores (i.e., predictors of competence), linear regression models were run, modeling baseline EBPAS-36, SU-SMS, AOS, and SDS vignette total scores, IPIP-NEO-60 conscientiousness and openness subscale scores, and total years of PRS work experience as predictors.”

  1. Tables: A table for role-play coding i.e., based on type of questions, responses, or theme. A table for a list of questions used in the interactive discussions.

RESPONSE: We have now clarified the role play instructions given to the RAs and participants in the methods section (as noted above). We felt as though describing this in-text rather than in a table is clearest, because there was not a specific script followed by the RAs during the role plays. We do agree that it would be helpful to include the codebook, and now include this as Table 1.

I enjoyed reading your paper. It is prepared with detailed explanations. It will be better if you can relate both theoretical and practical contributions of your study in the discussion part to enhance your research novelty since I believe many PhD students worldwide will read and benefit from your work. Anyway, the paper is ready for publication as it is.

RESPONSE: Thank you very much – we appreciate your thoughtful feedback. In response to Reviewer’s 2 feedback, we now describe the preliminary nature of this study and how these preliminary results may inform future research in the discussion (see #14 below).

Reviewer 2 Report

Thank you for opportunity for reviewing this paper. The topic of the manuscript is important area for good quality studies. The research area is indeed very interesting for a potential reader. Moreover, the added value is due to a potential practical application of the study results.

I believe that this manuscript doesn´t qualify for acceptance at this time and should be improved for publication. Than, it may be considered for acceptance.

Specific comments:

1.      Writing

The English used is appropriate.

 2.      Title

The title reflects the content and problem studied. The title should reflect what type of study is presented.

Moreover, geographical coverage of the research should be mentioned – was it done in one state, multiple states, country-wide? The country or state should be mentioned in title.

3.      Abstract

The abstract reflects the manuscript and provide a summary of what was done and what was found.

What may be found confusing is the frequent usage of the abbreviations and acronyms.

Like in the title, geographical location of the respondents should be mentioned.

4.    Key Words

The keywords are representative of the subject of the study.

5.      Introduction

The introduction reflects the state of the art in relation to the study. Nonetheless, the literature citied is not sufficient. Please add additional sources for the presented research results or statements, as currently only one source is used to support statement(s).

Moreover, IJERPH is a world-wide coverage journal, therefore it should be clearly stated if statements refer only to the USA. For example, in the statement: “In 2020 alone, 37.1 million people needed, yet did not receive, SUD treatment” – it should be mentioned concerning  people in the USA.

Also, the SUD abbreviation is used without full phrase usage.

Another strong statement “This treatment gap has differentially impacted underserved, ethnic/racial minoritized populations” – may indicate that treatment gap affects only ethnic/racial minoritized populations. If it so in the USA? Are PRSs only aimed to help ethnic/racial minoritized populations or to all groups in need, like socially disadvantaged, peoples with disabilities, despite ethnic or racial background? If no, more sources should be cited to support the statement.

6. Materials and Methods

Participants – what I found missing is again geographical area in relation to the recrutation – single state or multiple ones. If so, which ones?

The PRS certification is mentioned. As I understand, it refers to a certification process in the USA. Potential readers from, for example Europe, Asia, etc. may not be familiar about the certification – I suggest adding explanation. What type of certification? What institution(s) certify? In what countries or states it is valid?

The snowball method of recrutation is mentioned. What I found interesting is how many and what type of institutions the 20 PRSs represented?

REDCap is mentioned. Please also specify how many items were used, what types of questions, what groups of question topics were used in the assessment.

MHA estimates there are over 30,000 peer supporters working around the US (https://www.mhanational.org/peer-workforce). Nonetheless, the authors included only a very small sample of respondents. No justification in the text is provided why such a sample size was applied. In my opinion it should be clarified in the manuscript. Even as a pilot study, such a small sample may be insufficient, even if recruitation was done in a single state.

7. Results

I found this part very short. Nonetheless, I understand the authors due to the small sample.

8. Discussion

As mentioned in the introduction, underserved, ethnic/racial minoritized populations are especially suited to be supported by PRSs. What I found missing in the section is deepening the area. Do other studies point out what areas of such trainings in PRS competences and BA may be especially developed in order to best serve the target.

9. References

The references are used correctly although there are many new references that could be used, especially on in the introduction and discussion related sections.

Author Response

Thank you for opportunity for reviewing this paper. The topic of the manuscript is important area for good quality studies. The research area is indeed very interesting for a potential reader. Moreover, the added value is due to a potential practical application of the study results.

I believe that this manuscript doesn´t qualify for acceptance at this time and should be improved for publication. Then, it may be considered for acceptance.

RESPONSE: Thank you for your review of our manuscript and for highlighting the importance of this work. We have responded to your feedback below and believe the manuscript has been strengthened as a result.

Specific comments:

Writing

  1. The English used is appropriate.

RESPONSE: Thank you.

Title

  1. The title reflects the content and problem studied. The title should reflect what type of study is presented. Moreover, geographical coverage of the research should be mentioned – was it done in one state, multiple states, country-wide? The country or state should be mentioned in title.

RESPONSE: Thank you for raising these points. We agree that it would be beneficial to include the geographic location and study type in the title, and have edited the title to the following: “A pilot study of training peer recovery specialists in behavioral activation in the United States.: preliminary outcomes and predictors of competence.” We chose to broadly mention the U.S. as the geographic location because we did not have any inclusion/exclusion criteria based on state, and participants did not fall neatly into any one specific portion of the U.S.  

Abstract

  1. The abstract reflects the manuscript and provide a summary of what was done and what was found. What may be found confusing is the frequent usage of the abbreviations and acronyms. Like in the title, geographical location of the respondents should be mentioned.

RESPONSE: We have edited the abstract to decrease the number of acronyms and now include the geographical location of the participants.

Key Words

  1. The keywords are representative of the subject of the study.

 RESPONSE: Thank you.

Introduction

  1. The introduction reflects the state of the art in relation to the study. Nonetheless, the literature citied is not sufficient. Please add additional sources for the presented research results or statements, as currently only one source is used to support statement(s).

RESPONSE: Thank you for raising this point. We have now edited the paper to increase the number of sources cited throughout. For example, we have added additional citations to support the following sentences (though this is not an exhaustive list):

“This treatment gap has disproportionately impacted underserved, ethnic/racial minoritized populations compared to non-racial/ethnically minoritized populations [2,3].” (pg. 3)

“BA has been found to be feasible and acceptable for PRS delivery in historically underserved, racial/ethnically minoritized communities to increase engagement in SUD treatment [29], including MT retention [30,31], and PRS-delivered BA has been found to be effective in supporting MT retention [6].” (pg. 13)

  1. Moreover, IJERPH is a world-wide coverage journal, therefore it should be clearly stated if statements refer only to the USA. For example, in the statement: “In 2020 alone, 37.1 million people needed, yet did not receive, SUD treatment” – it should be mentioned concerning  people in the USA. Also, the SUD abbreviation is used without full phrase usage.

RESPONSE: We have edited to now make sure it is clear when we are referring to the U.S. specifically, and that all acronyms are properly cited at first use.

  1. Another strong statement “This treatment gap has differentially impacted underserved, ethnic/racial minoritized populations” – may indicate that treatment gap affects only ethnic/racial minoritized populations. If it so in the USA? Are PRSs only aimed to help ethnic/racial minoritized populations or to all groups in need, like socially disadvantaged, peoples with disabilities, despite ethnic or racial background? If no, more sources should be cited to support the statement.

RESPONSE: The statement, “This treatment gap has differentially impacted underserved, ethnic/racial minoritized populations,” is not to say that White individuals do not experience the negative impacts of the treatment gap or experience difficulty in accessing treatment, but that these impacts disproportionately impact ethnic/racial minoritized populations at a higher rate than their White counterparts. PRS services are not exclusively for ethnic/racially minoritized populations, but have rather been utilized to increase access to care in low-resource settings. We have clarified both of these points and believe that the statements are now clearer:

“In 2020 alone, 37.1 million people in the U.S. needed, yet did not receive, substance use disorder (SUD) treatment (1). This treatment gap has disproportionately impacted underserved, ethnic/racial minoritized populations compared to non-racial/ethnically minoritized populations [2]. To increase access to care in low-resource, underserved communities, the workforce of peer recovery specialists (PRSs), individuals with lived substance use and recovery experience, has rapidly expanded [3].”

Materials and Methods

  1. Participants – what I found missing is again geographical area in relation to the recrutation – single state or multiple ones. If so, which ones?

RESPONSE: Thank you for raising this point. As noted above, we did not have any inclusion/exclusion criteria around geographic location, which led to participants being from various states in the U.S. (primarily the DC-Maryland-Virginia area, which we include as a limitation). We have now included this in the methods.

“There were no exclusion criteria regarding geographic location. Participants were from various states in the U.S., primarily (though not limited to) the DC-Maryland-Virginia area, where the research team is located.”

  1. The PRS certification is mentioned. As I understand, it refers to a certification process in the USA. Potential readers from, for example Europe, Asia, etc. may not be familiar about the certification – I suggest adding explanation. What type of certification? What institution(s) certify? In what countries or states it is valid?

RESPONSE: In the U.S., PRS certification requirements vary state-by-state. We now include this information, along with a citation that details the requirements per state, in the methods:

Of note, PRS certification requirements in the U.S. vary by state , and may include: years/time in recovery, service hours, hours spent being supervised, examination(s), and ongoing continued education credits [22]. Many states utilize the Connecticut Community for Addiction Recovery (CCAR) approved training materials which often include domains such as: ethics, advocacy, mentoring, and recovery/wellness [22].”

  1. The snowball method of recrutation is mentioned. What I found interesting is how many and what type of institutions the 20 PRSs represented?

RESPONSE: We agree that it would be valuable to include this information. We now provide a list of types of institutions in the results along with other participant demographic information:

Participants worked in a variety of settings/roles, including: community outreach, non-governmental organizations/non-profits (including shelters), departments of health, and public substance use treatment programs (including clinics and hospitals).”

  1. REDCap is mentioned. Please also specify how many items were used, what types of questions, what groups of question topics were used in the assessment.

RESPONSE: The measures that are listed in the Measures section of the Methods (i.e., EBPAS-36, SU-SMS, AOS, SDS and IPIP-NEO-60) were those that were administered via REDCap. We have now clarified this at the first mention of REDCap (p. 7).

“Assessments included both written quantitative measures (detailed below) completed independently by participants (with RA assistance if needed) using an online data collection tool, REDCap, as well as a role play with a trained RA.”

  1. MHA estimates there are over 30,000 peer supporters working around the US (https://www.mhanational.org/peer-workforce). Nonetheless, the authors included only a very small sample of respondents. No justification in the text is provided why such a sample size was applied. In my opinion it should be clarified in the manuscript. Even as a pilot study, such a small sample may be insufficient, even if recruitation was done in a single state.

RESPONSE: We appreciate the point that the reviewer is raising. We note that this is an initial pilot study that is intended to guide the development of a future, large-scale study and, to that end, includes a smaller sample size. We have endeavored to clarify that this is a pilot and noted in the limitations section that the small sample size limits the generalizability of these findings:

“As this study reports on a small sample size (N = 20), due to the preliminary nature of the study, the ability to detect predictors may have been limited. Given that this was a pilot study, future, larger-scale, studies should be conducted to replicate these results and explore additional predictors of post-training success.” (p. 14)

Results

  1. I found this part very short. Nonetheless, I understand the authors due to the small sample.

RESPONSE: We appreciate the reviewer’s comment, and believe that including more demographic information on the participants, as noted above, has also strengthened the content of this section.

Discussion

  1. As mentioned in the introduction, underserved, ethnic/racial minoritized populations are especially suited to be supported by PRSs. What I found missing in the section is deepening the area. Do other studies point out what areas of such trainings in PRS competences and BA may be especially developed in order to best serve the target.

RESPONSE: To the best of our knowledge, this is the first study with the focus of training PRSs in BA. However, we do cite research that found PRS-delivered BA to be feasible, acceptable and effective in low-resource settings among primarily ethnic-racially minoritized samples. This is now clarified in the discussion:  

 “BA has been found to be feasible and acceptable for PRS delivery in historically underserved, racial/ethnically minoritized communities to increase engagement in SUD treatment [29], including MT retention [30,31], and PRS-delivered BA has been found to be effective in supporting MT retention [6]. However, these studies utilize a single PRS interventionist, limiting the ability to understand what predicts competence in the delivery of the intervention. Thus, this was a pilot study to begin to understand what predictors may be associated with competence in this approach following a brief training.”

References

  1. The references are used correctly although there are many new references that could be used, especially on in the introduction and discussion related sections.

RESPONSE: We have now edited the paper to increase the number of sources cited throughout.
